# Carotenoids in Palliative Care—Is There Any Benefit from *Carotenoid* Supplementation in the Adjuvant Treatment of Cancer-Related Symptoms?

**DOI:** 10.3390/nu14153183

**Published:** 2022-08-03

**Authors:** Anna Zasowska-Nowak, Piotr Jan Nowak, Aleksandra Cialkowska-Rysz

**Affiliations:** 1Department of Palliative Medicine, Chair of Oncology, Medical University of Lodz, Hallera Square 1, 90-647 Lodz, Poland; aleksandra.cialkowska-rysz@umed.lodz.pl; 2Department of Nephrology, Hypertension and Kidney Transplantation, Medical University of Lodz, ul. Pomorska 251, 92-213 Lodz, Poland; piotr.nowak@umed.lodz.pl

**Keywords:** carotenoids, neuropathic pain, cancer cachexia, cancer-related fatigue, palliative care

## Abstract

Carotenoids are organic, liposoluble pigments found in nature, which are responsible for the characteristic colors of ripe tomatoes, carrots, peppers, and crustaceans, among others. Palliative care provided to patients with an incurable disease is aimed at improving the patient’s quality of life through appropriate treatment of symptoms accompanying the disease. Palliative care patients with burdensome symptoms related to advanced-stage cancers are especially interested in the use of natural dietary supplements and herbal remedies to reduce symptoms’ intensity and ameliorate the quality of life. Carotenoids seem to be a group of natural compounds with particularly promising properties in relieving symptoms, mainly due to their strong antioxidant, anti-inflammatory, and neuroprotective properties. Moreover, carotenoids have been used in folk medicine to treat various diseases and alleviate the accompanying symptoms. In this narrative review, the authors decided to determine whether there is any scientific evidence supporting the rationale for carotenoid supplementation in advanced-stage cancer patients, with particular emphasis on the adjuvant treatment of cancer-related symptoms, such as neuropathic pain and cancer-related cachexia.

## 1. Introduction

Carotenoids are organic, liposoluble pigments distributed in colored fruits, vegetables, photosynthetic bacteria, fungi, algae, and some fish [1]. Carotenoids are responsible for characteristic colors found in nature; for example, green leafy vegetables contain high amounts of xanthophylls and hydrocarbon carotenes, lycopene is present in ripe tomatoes, β-carotene is responsible for the orange color of carrots, capsanthin gives red color to peppers, and astaxanthin dyes crustaceans pink and red [1]. Carotenoids are not only substantial pigments involved in photosynthesis (along with chlorophylls) in photosynthetic bacteria, algae, and plants, but they also act as photoprotectors, color attractants, antioxidants, and precursors of plant-derived hormones [2]. Carotenoids are classified into two main groups, depending on their chemical structure and functions: xanthophylls (lutein, zeaxanthin, and astaxanthin) and carotenes (α-carotene, β-carotene, and lycopene) [3]. In humans and animals, who are not able to synthesize carotenoids de novo, these compounds need to be ingested with food or supplemented, as they play an important role as vitamin A precursors and bioactive compounds, which may participate in many physiological processes [2]. Although more than 850 carotenoids have been identified so far in the natural environment, only about 50 of them are found in the typical human diet [2]. However, only about 20 types of carotenoids have been found in human blood and tissues [4], among which β-carotene, α-carotene, lycopene, β-cryptoxanthin, lutein, and zeaxanthin are most often identified [2,4]. It has so far been demonstrated that carotenoids are able to perform many biological functions due to their antioxidant, anti-inflammatory, and anti-tumor activities; they also have the ability to inhibit angiogenesis, induction of apoptosis, cell cycle arrest, and enhance the immune response [1]. Furthermore, α-carotene, β-carotene, γ-carotene and β-cryptoxanthin possess pro-vitamin A activity, thus participating in the process of vision, and influencing the proper growth and development of bones and the proper functioning of the epithelium.

The consumption of a diet rich in fruits and vegetables significantly affects the risk for various chronic diseases. Donaldson [5], in a review of 62 studies of plasma carotenoids and health outcomes, indicated that a total plasma carotenoid level <1 µM is related to a very high risk of metabolic and cardiovascular diseases as well as cancer. On the other hand, a concentration of total plasma carotenoids >2.5 µM significantly influences the above risk, reducing it into a low level (a very low level occurs at >4 µM). Moreover, a meta-analysis conducted in 2021 by Hajizadeh-Sharafabad supported the possible protective effect of carotenoids on inflammatory biomarkers [6]. A particularly important, modifiable factor that positively influences plasma carotenoid concentration is a diet rich in fruit and vegetables. The Mediterranean diet is an excellent example of a properly balanced diet, the use of which has a positive effect on plasma carotenoid levels [7]. On the other hand, older women from neighborhoods with lower socioeconomic status have lower serum carotenoid concentrations, which reflect a lower consumption of carotenoid-rich fresh fruits and vegetables [8]. Unfortunately, the available literature data do not provide specific recommendations for carotenoid consumption in palliative care patients. 

Palliative care is an approach that improves the quality of life of patients (and their families) facing the burden of life-threatening illness through prevention and relief of suffering using early identification and adequate assessment and treatment of pain as well as other problems (physical, psychosocial, spiritual) [9]. Palliative care focused on improving quality of life is of particular importance in patients with advanced neoplastic disease refractory to anti-cancer therapy [10], especially since 82% of them have low quality of life scores, mostly resulting from burdensome symptoms [11]. Neuropathic pain and cancer cachexia are some of the most distressing symptoms among palliative cancer patients that negatively affect the quality of life, mainly due to their high prevalence, severity, and limited therapeutic options.

Although the scale of the use of diet supplements and herbal remedies among palliative-care patients remains unknown, it seems to enjoy unflagging interest from patients with advanced-stage cancer disease refractory to standard therapy. One of the most frequent questions asked by patients and their families to palliative medicine specialists concerns the effectiveness of various natural compounds, carotenoids among them, as a therapy that can improve prognosis or relieve refractory and distressing symptoms. 

In this narrative review, the authors deal with this issue, trying to determine whether there is any scientific evidence supporting the rationale for carotenoid supplementation in advanced-stage cancer patients, with particular emphasis on the adjuvant treatment of cancer-related symptoms. The chemical structures of the key carotenoids discussed in the manuscript are shown in Figure 1.

## 2. Carotenoids as an Adjuvant Treatment of Neuropathic Pain 

Neuropathic pain is defined as pain caused by lesion or disease of the somatosensory system, a part of the nervous system responsible for sensing touch, temperature, posture, and limb position, among others [12]. In patients with advanced cancer this special type of pain can be a direct consequence of a cancer-induced injury of the somatosensory system, or it can be triggered by the anti-cancer treatment, such as radiation therapy, chemotherapy, or surgery. In cancer patients, the overall prevalence of neuropathic pain ranges from 19% to 39% [13]. Clinical symptoms of neuropathic pain are quite characteristic and include pain sensations, such as increased sensitivity to painful stimuli (hyperalgesia), the perception of innoxious stimuli as painful (allodynia), and spontaneous pain [14]. Such unpleasant sensations are the reason for the poor quality of life observed in patients with advanced cancers. Based on the results of clinical trials conducted so far, it has been concluded that the most effective form of neuropathic pain therapy is the use of strong opioid analgesics or tramadol in combination with calcium channel a2-d ligands (gabapentin, pregabalin) or certain antidepressants (tricyclic antidepressants and dual reuptake inhibitors of both serotonin and norepinephrine) [15]. Despite efforts to provide a good control of neuropathic pain, almost one-third of cancer patients do not achieve a satisfactory analgesic effect [16]. Moreover, the abovementioned treatment is associated with a high risk of side effects, whose occurrence can lead to a reduction of in the doses of painkillers or even to the discontinuation of analgesic treatment. Therefore, the search for novel therapeutic strategies effective in the treatment of neuropathic pain still remains a challenge. Natural compounds and herbal medicines that has been used for centuries to reduce pain intensity and inflammation are of great interest nowadays [17]. Considering the pathophysiology of neuropathic pain, it can be hypothesized that chemical compounds such as carotenoids with strong anti-inflammatory and antioxidant properties may be of particular importance in the alleviation of neuropathic pain. 

A special role in the pathogenesis of chronic neuropathic pain is played by the cascade of immune responses and oxidative stress induced by nerve injury [14]. While single, acute pain stimuli disappear and do not have any serious consequences, repeated stimuli quite quickly cause adaptive changes in both the peripheral and central nervous systems. Endothelial damage, an increased neuronal activity, and several mechanisms that modify pain perception are activated both at the site of nerve damage, in the dorsal root ganglia (DRG), and in the dorsal horn of the spinal cord. The recruitment of monocytes/monophages in the proximity of the damaged nerve and central activation of non-neuronal cells, such as microglia in the brain and spinal cord, lead to a subsequent release of proinflammatory mediators, resulting in both peripheral and central sensitizations that enable positive feedback and maintaining chronic pain [18]. Endothelial activation at the site of nerve injury results in recruitment of CX3C chemokine receptor 1 (CX3CR1)-expressing monocytes/macrophages, what in turn sensitize nociceptive neurons through the release of reactive oxygen species (ROS) that activate the transient receptor potential ankyrin 1 (TRPA1) channel to evoke a pain response [18]. On the other hand, spinal glia cells that are activated as a result of nerve damage release neuromodulators, such as proinflammatory cytokines interleukin-1 beta (IL-1β) and interleukin-6 (IL-6), tumor necrosis factor-alpha (TNF-α), growth factors, and chemokines responsible for the intensification and preservation of neuropathic pain [14]. In astrocytic and microglial activation the overactivity of glutamatergic N-methyl-D-aspartate (NMDA) receptors also participates [19]. Neuroinflammation and oxidative stress are considered to be factors implicated in the pathogenesis of neuropathic pain through sensitization of the nociceptive receptors, among others [14]. 

Thus, carotenoids with both anti-inflammatory and antioxidant properties seem to be an excellent example of naturally occurring active substances capable of alleviating neuropathic pain. Additional carotenoid supplementation in parallel with standard analgesic treatment may not only contribute to better pain control and alleviation of side effects of treatment, but also due to additional properties may improve other inflammation-related symptoms, such as cancer-related cachexia, anorexia, or asthenia. In the following subsections, the authors will try to determine whether the results of the studies conducted so far support the benefits of additional supplementation of carotenoids such as crocetin/crocin, astaxanthin, β-cryptoxanthin, fucoxanthin, bixin, and lycopene in the treatment of neuropathic pain.

### 2.1. Crocetin/Crocin

Crocetin and its glycosidic ester crocin are the water-soluble carotenoids responsible for the unique color of saffron, a spice sourced from the dried stigmas of *Crocus sativus* L. Saffron extracts and tinctures have been used for ages in traditional medicine for the treatment of different symptoms and diseases, due to its antispasmodic, sedative, diaphoretic, expectorant, and analgesic properties [20]. Saffron and its pharmacologically active compound crocin have been shown to be a powerful antioxidant and anti-inflammatory agent [20]. In vitro experiments and studies conducted in animal models have shown that crocin not only neutralizes ROS, but also is able to suppress the secretion of proinflammatory cytokines and mitigates inflammation in various organs (such as lung, heart, brain, and kidneys), mainly via regulation of the nuclear factor kappa-light-chain-enhancer of activated B cells (NF-κB) pathway [21]. What is more, recent studies with both in vitro and in vivo models indicate that saffron compounds, particularly crocin and crocetin, may have an anticancer effect in breast, lung, pancreatic, and leukemic cells [22]. 

As mentioned above, neuroinflammation and oxidative stress have been identified as factors playing a crucial role in the pathogenesis of neuropathic pain [14]. Data obtained from experimental studies demonstrate that centrally administered crocetin and its by-product crocin can affect pain-regulating mechanisms, presumably due to their anti-inflammatory, anti-oxidative, and neuroprotective properties [14]. It is worth mentioning that crocin is not systemically detected after oral administration, as it is hydrolyzed by enzymes of the intestinal epithelium and microbiota to deglycosylated trans-crocetin, absorbed passively through the intestinal mucosa layer [20]. As has been shown in studies on laboratory animals, trans-crocetin is the only metabolite capable of penetrating the blood-brain barrier [20]. 

Experimental studies conducted on animal models of neuropathic pain have shown that parenterally and centrally administered crocetin and crocin possess the ability to alleviate pain sensitization; however, no consistent data on the analgesic mechanism have been established. The results of a study by Erfanparast et al. [23] suggested that central H_2_ histaminergic and alpha-2 adrenergic receptors might be involved in crocetin-induced antinociception. Wang et al. [24] proposed that crocin may alleviate neuropathic pain by inhibiting the production of proinflammatory molecules TNF-*α* and IL-1*β* through the Wnt5a/*β*-catenin pathway, as Wnt Family Member 5A (Wnt5a) plays an important role in central sensitization and neuronal plasticity in both acute and chronic pain. On the other hand, Vafaei et al. [25] suggested that the analgesic effect of crocin is probably mediated by an endocannabinoid mechanism, whereas Safakhah et al. [26] postulated an important role for muscarinic receptors in the antinociceptive effect of crocin. Tamaddonfard et al. [27] in turn ruled out the action of crocetin through the opioid system.

In a mouse model of neuropathic pain induced by spared nerve injury (SNI), intrathecally perfused crocetin dose-dependently reduced mechanical and thermal allodynia [14]. Moreover, repeated treatment with high doses of crocetin (30 mg/kg) prevented SNI-induced increase of TNF-α and IL-1β and restored reduced activity of mitochondrial superoxide dismutase (MnSOD) in the sciatic nerves and spinal cords of laboratory mice [14]. In addition, in a study by Erfanparast et al. [23], crocetin administered into the cerebral fourth ventricle alleviated formalin-induced orofacial pain in laboratory rats, although the doses used were many times lower than in the abovementioned study by Wang et al. [14] (5 and 10 µg). What is important is that crocetin reduced both the first transient phase of pain behavior, associated with direct effect of formalin on sensory C fibers, as well as the second, prolonged tonic pain phase, related to peripheral inflammatory response and central sensitization [23]. 

Centrally administered crocin at a dose of 6 µg significantly decreased thermal hyperalgesia and mechanical allodynia in a chronic constriction injury (CCI)-induced model of neuropathic pain in laboratory rats [25]. Intraperitoneally (i.p.)-administered crocin at a dose of 60 mg/kg in a study by Safakhah et al. [26] in the same model of neuropathic pain significantly reduced both mechanical allodynia and thermal hyperalgesia. In another experimental study, i.p. injection of crocin in the same dose significantly decreased mechanical allodynia, but not thermal analgesia [25]. However, i.p. administration at lower doses (e.g., 50 mg/kg) did not show any antinociceptive effect in a CCI–induced model of neuropathic pain [28]. 

A neuroprotective effect of crocin was also observed in the streptozocin (STZ)-induced model of diabetic neuropathy. Farshid and Tamaddonfard [29] proved that (i.p.)-administered crocin at the dose of 30 mg/kg improved not only STZ-induced cold allodynia and hyperglycemia, but histopathological degenerative changes of the sciatic nerve and elevation of sciatic nerve malondialdehyde (MDA) levels. 

The analgesic effect induced by centrally and peripherally administered crocin has also been demonstrated in other pain models. Tamaddonfard et al. [30] showed that the injection of crocin into the cerebral fourth ventricle at doses of 10 and 40 µg attenuated capsaicin-induced orofacial pain in laboratory rats. A dose-dependent significant antinociceptive effect of peripherally administered crocin (50, 100, and 200 mg/kg i.p.) was observed in the formalin-induced model of pain [27]. In the carrageenan-induced model of local inflammation and inflammatory pain, i.p.-administered crocin (25, 50, and 100 mg/kg) attenuated edema, suppressed pain responses, and decreased the number of neutrophils infiltrating the site of carrageenan application [31]. 

As shown above, crocin has been demonstrated to attenuate neuropathic pain in animal models (Table 1). In the only randomized, double-blind, placebo-controlled trial conducted so far by Bozorgi et al. [32], its statistically significant effectiveness for relieving symptoms of chemotherapy-induced peripheral neuropathy was confirmed. In that study, cancer patients were randomly treated with 15 mg crocin twice daily by oral route versus placebo for 8 weeks (details of this study are presented in Table 2). During the study it was also observed that crocin had no serious adverse effects, except nausea (grade 2). The incidence of other adverse effects did not differ statistically significantly between the study and the control groups, and included increased appetite, sedation, headache, hypomania, stomachache, vomiting, and swelling of the feet [32]. 

### 2.2. Astaxanthin

Astaxanthin is considered to be the carotenoid with the strongest antioxidant capacity when compared to other carotenoids (such as zeaxanthin, lutein, tunaxanthin, canthaxanthin and β-carotene) and commonly used antioxidants, such as α-tocopherol (vitamin E) [33,34]. This red-orange natural pigment is derived from β-carotene by 3-hydroxylation and 4-ketolation, and belongs to a group of carotenoids called xanthophylls that includes, in addition to astaxanthin, such chemical compounds as β-cryptoxanthin, β-carotene, lycopene, and lutein/zeaxanthin [35,36]. Its C40 molecular structure is similar to β-carotene, but the possession of 13 conjugated double polyunsaturated bonds gives astaxanthin unique chemical properties, makes its molecule more polar, and enhances its antioxidant capacity [36]. Both lipophilic and hydrophilic properties of the astaxanthin molecule make it unique compared to other carotenoid antioxidants, such as β-carotene, as it is exposed to both the inside and outside of the cell, scavenging radicals both from surface of the cell and the interior of the phospholipid membrane [34]. Moreover, as it is fat-soluble astaxanthin can pass the blood-brain barrier [33]. 

Marine microalgae such as *Haematococcus pluvialis* and *Chlorella vulgaris* and the red yeast *Phaffia rhodozyma* are considered rich sources of astaxanthin. However, it also can be ingested with crustaceans (copepods, shrimp, and krill) and *Salmonidae* species (salmon, rainbow trout), containing natural sources of astaxanthin in their diet [34]. 

In recent years astaxanthin has been widely studied in in vitro tests, in animal models, and in human clinical trials, due to its antioxidant, anti-inflammatory, anti-apoptotic, neuroprotective, cardioprotective, hepatoprotective, anti-cancer, anti-diabetic, and immuno-modulatory properties [34,35]. Astaxanthin has also been evaluated in the treatment of neuropathic pain, and it was proposed that neuroprotection mediated by astaxanthin results from its anti-oxidative, anti-inflammatory, and anti-apoptotic mechanisms [34]. Fakhri et al. [35] proposed that in neuropathic pain following spinal cord injury (SCI), astaxanthin reduced mechanical allodynia by blocking NMDA receptor subunit 2B (NR2B), and by the inhibition of macrophage migration inhibitory factor (MIF), protein kinase B (AkT), phosphorylated p38 mitogen-activated protein kinase (p-p38MAPK), and extracellular signal-regulated kinase (p-ERK), and stimulation of the phosphorylated form of protein kinase B (p-AkT) and extracellular signal-regulated kinase (ERK) pathways. However, despite some hypotheses that have been presented on the basis of the results of the obtained data, a clear mechanism responsible for the analgesic properties of astaxanthin in neuropathic pain has not been identified. 

In the promising study by Zhao et al. [33], astaxanthin alleviated neuropathic pain in a spinal nerve ligation (SNL)-induced mouse model of neuropathic pain. Its peritoneal administration (5 and 10 mg/kg) significantly inhibited mechanical allodynia and thermal hyperalgesia in a dose-dependent manner. Moreover, it was shown that astaxanthin inhibited the activation of microglia cells, astrocytes, and neurons, inhibited the expression of proinflammatory cytokines (IL-1β, IL-6, IL-17, and TNF-α) and phosphorylation of inflammatory signaling mediators (ERK1/2, p38 mitogen-activated protein kinase (p38 MAPK)), inhibited NF-κB p65 activation, and simultaneously increased the secretion of the anti-inflammatory interleukin-10 (IL-10) in spinal dorsal horn tissue and cultured cells [33]. In a study by Masoudi et al. [37], astaxanthin treatment not only alleviated SCI-induced neuropathic pain in a mouse model, but also reduced oxidative stress and expression levels of cyclooxygenase-2 (COX-2), TNF-α, IL-1β, and IL-6. Furthermore, histopathological evaluation revealed that myelinated white matter and motor neuron numbers were significantly preserved beyond the injury after astaxanthin treatment [37]. Fakhri et al. [38] proved that astaxanthin decreased the expression of inflammatory signaling mediators, p-p38MAPK, and inflammatory cytokine TNF-α in the severe compression model of SCI in male rats. Histopathological examination showed that astaxanthin was able to preserve the white matter and motor neurons beyond the injury in rostral and caudal parts [38]. On the other hand, Sharma et al. [19] in an in silico molecular docking study ascertained the binding affinity of astaxanthin to NMDA receptors and showed its antagonistic effect. Moreover, in in vitro study by the same authors astaxanthin significantly reduced lipopolysaccharide (LPS)-induced oxidative stress in C6 glial cells, whereas laboratory rats presented significant attenuation of neuropathic pain behavior after astaxanthin treatment [19]. Mohammadi et al. [39], in turn proposed that the antinociceptive activity of astaxanthin is related to the signaling pathway of nitric oxide (NO), 3′, 5′-cyclic guanosine monophosphate (cGMP), and ATP-sensitive potassium (K^+^) channels (L-arginine/NO/cGMP/K_ATP_) playing the modulatory role in the perception of pain. 

Moreover, it has been proven that the trans-isomer of astaxanthin exerts an anti-depressant-like effect by influencing the serotonergic system, and it may be of particular interest in the treatment of chronic neuropathic pain in cases with pain-related depressive disorders coexisting [40]. In the preclinical study conducted by Jiang et al. [40], chronic trans-astaxanthin treatment not only ameliorated mechanical allodynia and thermal hyperalgesia in a CCI mouse model, but it was also able to reverse increased indoleamine 2,3-dioxygenase (IDO) expression in the hippocampus and spinal cord and the kynurenine (KYN)/tryptophan (TRY) ratio, as well as decrease 5-hydroxytryptamine (5-HT)/TRY and 5-HT/5-hydroxyindoleacetic acid (5-HIAA) ratios as the result of CCI surgery. Moreover, trans-astaxanthin at a daily dose of 80 mg/kg effectively antagonized IL-1β, IL-6, and TNF-α expression in the hippocampus and spinal cords of CCI mice [40]. 

Unfortunately, the authors did not find in the literature data from human studies that would confirm the analgesic efficacy of astaxanthin. Despite the promising results of studies conducted so far in silico, in vitro, and in vivo on animal models, it is impossible to properly assess the usefulness of astaxanthin in the treatment of neuropathic pain in cancer patients. Further research in humans should be conducted in this area. A summary of preclinical studies on the antinociceptive effect of astaxanthin on animal models of neuropathic pain is presented in Table 1. 

### 2.3. Other Xanthophylls: β-Cryptoxanthin and Fucoxanthin 

The only experimental study to evaluate the use of β-cryptoxanthin, a major xanthophyll carotenoid routinely found in human serum, in the relief of symptoms associated with neuropathic pain was carried out by Park et al. [41] on a mouse model with spinal nerve injury. In that study the authors found that oral administration of β-cryptoxanthin in a daily dose of 10 mg/L for 28 consecutive days significantly reduced the development of tactile allodynia following spinal nerve injury. However, that procedure turned out to be ineffective in mechanical allodynia and in inflammatory pain in laboratory mice [41]. 

Fucoxanthin is a carotenoid with many pharmaceutical properties that is found in brown seaweed [42]. Chen et al. [42] in a preclinical study on ultraviolet B (UVB)-induced photokeratitis using an animal model observed the protective role of fucoxanthin on ocular lesions, such as corneal denervation and trigeminal pain. Moreover, the active phosphorylated form of p38 MAP kinase (p-p38 MAPK) and the number of glial fibrillary acidic protein (GFAP)-positive neural cells were significantly reduced, and decreased expression of neuron-selective transient receptor potential vanilloid type 1 (TRPV1) in the trigeminal ganglia neurons was demonstrated in rats treated with fucoxanthin after UVB-induced keratitis [42].

As shown above, the analgesic properties of both β-cryptoxanthin and fucoxanthin have not been tested in clinical trials. In addition, studies on animal models are few, and so far do not allow for consistent conclusions about the analgesic properties of these xanthophylls. A summary of preclinical studies on the antinociceptive effect of β-cryptoxanthin and fucoxanthin on animal models of neuropathic pain is presented in Table 1. 

### 2.4. Bixin 

Bixin is a natural apocarotenoid extracted from *Bixa orellana* seeds. This natural pigment is commonly used as a cosmetic and textile colorant and in the food industry as well, in the coloring of products such as margarine, yellow cheese, etc. The potential anti-inflammatory and antinociceptive effects of bixin in daily doses of 15 or 30 mg/kg body mass orally were investigated in a preclinical model of inflammation and acute pain in a study by Pacheco et al. [43]. In that study it was observed that only the higher dose of bixin (30 mg/kg) significantly decreased carrageenan-induced paw edema and MPO activity, and increased the latency time in the hot plate test. However, both doses of bixin positively influenced the results of both phases of the formalin test, without affecting locomotor performance [43]. In the experimental model of diabetes Gasparin et al. [44] observed that treatment with bixin in daily doses of 30 or 90 mg/kg orally repeated for 17 days significantly attenuated mechanical allodynia and depressive and anxious behaviors in STZ-induced diabetic rats without changing locomotor performance. Moreover, it was observed that bixin normalized oxidative parameters in tissues and reduced the plasma glycated hemoglobin A1 (HbA1). The authors of the abovementioned studies underlined that further studies are necessary to characterize the mechanisms involved in the analgesic effect of bixin and to evaluate its analgesic effectiveness in human studies. 

### 2.5. Lycopene 

In a partial sciatic nerve ligation-induced mouse model of neuropathic pain, Zhang et al. [45] observed that repeated intrathecal administration of lycopene prevented the occurrence of mechanical hypersensitivity. Moreover, the authors of the study proposed that the most significant mechanism that mediated the analgesic effect of lycopene is the upregulation of spinal astrocytic connexin 43 (Cx43) expression [45]. Beneficial properties in the alleviation of symptoms associated with peripheral neuropathic pain, such as thermal hyperalgesia and cold allodynia, were also observed in an animal model of STZ-induced diabetic neuropathy [46]. Study authors Kuhad and Chopra [46] proved that chronic treatment of lycopene in a daily dose of 4 mg/kg body weight orally for 4 weeks significantly reduced the abovementioned sensory experiences. It was proposed that the antinociceptive effect of lycopene may be attributed to its antioxidant activity and vasorelaxant properties, leading to improvement in neuronal blood flow [46]. Kuhad et al. [47], in another experimental study on STZ-induced diabetic neuropathy in laboratory mice, observed that lower doses of lycopene (1, 2, and 4 mg/kg body weight) also significantly attenuated thermal hyperalgesia, while inhibiting the TNF-α and NO release in a dose-dependent manner. A summary of preclinical studies on the antinociceptive effect of lycopene in animal models of neuropathic pain is presented in Table 1. 

The authors identified only two human intervention studies that tested the effectiveness of lycopene supplementation on cancer-related pain. However, none of these studies have evaluated the effectiveness of lycopene in relieving neuropathic pain. In the clinical trial conducted in 2003 by Ansari and Gupta [48], patients taking lycopene in a dose of 2 mg twice a day simultaneously with basic treatment of metastatic prostate cancer (orchidectomy) used analgesics less frequently than the control group (orchidectomy alone) (15 vs. 25%). In another clinical trial published by the same authors in 2004, lycopene in a daily dose of 10 mg was administered for 3 months in patients with metastatic hormone-resistant prostate cancer [49]. After that treatment it was observed that bone pain was significantly reduced, which allowed the reduction of analgesics in 62% of studied patients [49]. Details of these studies are presented in Table 2.

**Table 1 nutrients-14-03183-t001:** Preclinical studies on the antinociceptive effect of carotenoids on animal models of neuropathic pain.

Carotenoid	Study Description	Main Results	References
Crocetin	SNI-induced neuropathic pain in mice; crocetin administered intrathecally at doses of 5–50 mg/kg body mass for up to 12 days starting 3 days before the surgery.	Alleviation of mechanical and thermal allodynia in a dose-dependent manner. Reduction of SNI-induced increased levels of TNF-α and IL-1β. Restoration of SNI-induced reduction of MnSOD level in the sciatic nerve and the spinal cord.	Wang et al. (2017) [14].
Crocetin	Formalin-induced orofacial pain in laboratory rats; crocetin administered into the cerebral fourth ventricle at doses of 2.5, 5 and 10 μg.	Crocetin at doses 5 and 10 μg significantly attenuated the first and the second phases of formalin-induced orofacial pain.	Erfanparast et al. (2020) [23]
Crocin	CCI-induced neuropathic pain in male rats; crocin administered intracerebroventricularly at dose of 6 µg or intraperitoneally at a dose of 60 µg/kg.	Centrally administered crocin significantly decreased thermal hyperalgesia and mechanical allodynia. Peripheral injection significantly decreased mechanical allodynia but not thermal hyperalgesia.	Vafaei et al. (2020) [25]
Crocin	CCI-induced neuropathic pain in male rats; crocin administered intraperitoneally at a dose of 60 µg/kg.	Significantly decreased thermal hyperalgesia and mechanical allodynia.	Safakhah et al. (2020) [26]
Crocin	CCI-induced neuropathic pain in male rats; crocin administered intraperitoneally at doses of 12.5, 25 and 50 mg/kg.	No analgesic effect.	Amin et al. (2012) [28]
Crocin	STZ-induced model of diabetic neuropathy; crocin administered intraperitoneally at a dose of 30 mg/kg	Alleviation of thermal allodynia. Reduced histopathological degenerative changes of sciatic nerve. Restoration of STZ-induced reduction of MDA level in the sciatic nerve.	Farshid and Tamaddonfard (2015) [29]
Crocin	Capsaicin-induced orofacial pain in male rats; crocin administered intracerebroventricularly at doses of 2.5, 10 and 40 µg.	Crocin at doses 10 and 40 μg significantly decreased severity of pain.	Tamaddonfard et al. (2015) [30]
Crocin	Formalin-induced pain in rats; crocin administered intraperitoneally at doses of 50, 100 and 200 mg/kg.	Significant reduction of pain. Crocin at a dose of 100 mg/kg significantly increased morphine-induced antinociception. Crocin at a dose of 400 mg/kg significantly suppressed locomotor activities.	Tamaddonfard et Hamzeh-Gooshchi (2010) [27]
Crocin	Carrageenan-induced inflammatory pain in male rats; crocin administered intraperitoneally at doses of 25, 50 and 100 mg/kg.	Reduced pain responses. Attenuated edema. Decreased number of neutrophils infiltrated the site of carrageenan application.	Tamaddonfard et al. (2013) [31]
Astaxanthin	SNL-induced neuropathic pain in mice; astaxanthin administered intraperitoneally at doses of 5 and 10 mg/kg.	Significant alleviation of mechanical allodynia and thermal hyperalgesia in a dose-dependent manner. Decreased: - expression of IL-1β, IL-6, IL-17, TNFα, - phosphorylation of ERK1/2 and p38 MAPK, - activation of NF-κB p65 and - increased secretion of IL-10 in the spinal dorsal horn cells.	Zhao et al. (2021) [33]
Astaxanthin	SCI-induced neuropathic pain in laboratory mice.	Significant alleviation of mechanical allodynia. Decreased expression of COX-2, TNFα, IL-1β, and IL-6. Reduction in the level of oxidative stress. Histopathologically confirmed protective effect against SCI-induced injury of white matter and motor neurons.	Masoudi et al. (2021) [37]
Astaxanthin	SCI-induced neuropathic pain in male rats; 10 µL of 0.2 mM astaxanthin solution administered intrathecally.	Decreased expression of TNF-α, p-p38 MAPK and NMDA receptor NR2B subunit in spinal cord. Histopathologically confirmed protective effect against SCI-induced injury of white matter and motor neurons.	Fakhri et al. (2018) [38]
Astaxanthin	CCI-induced neuropathic pain in male rats; astaxanthin administered intraperitoneally at doses of 5 and 10 mg/kg.	Significant attenuation of mechanical allodynia and thermal hyperalgesia.	Sharma et al. (2018) [19]
Trans-astaxanthin	CCI-induced neuropathic pain in male mice; trans-astaxanthin administered orally at doses of 10, 40 and 80 mg/kg twice a day, began 7 days after the surgical procedure and continued for 3 weeks.	Ameliorated mechanical allodynia and thermal hyperalgesia. Reversed CCI-induced increase of IDO expression and KYN/TRY ratio and decreased 5-HT)/TRY and 5-HT/5-HIAA ratios in hippocampus and spinal cord. Trans-astaxanthin at dose of 80mg/kg reduced IL-1β, IL-6 and TNF-α expression in hippocampus and spinal cord.	Jiang et al. (2018) [40]
Beta-cryptoxanthin	SNI-induced neuropathic pain in male mice; β-cryptoxanthin administered orally at a daily dose of 10 mg for 28 consecutive days.	Significant reduction of tactile allodynia.	Park et al. (2017) [41]
Fucoxanthin	UVB-induced trigeminal pain in rats; fucoxanthin administered orally at doses of 0.1, 1 and 10 mg/kg for 6 days.	Reduction of p-p38 MAPK and TRPV1 expression in trigeminal ganglia neurons. Decreased number of GFAP-positive neural cells in trigeminal ganglia.	Chen et al. (2019) [42]
Bixin	STZ-induced neuropathic pain in laboratory mice; bixin administered orally at doses of 10, 30 and 90 mg/kg for 17 days.	Bixin in doses of 30 and 90 mg/kg significantly alleviated mechanical allodynia and anxious behaviors.	Gasparin et al. (2021) [44]
Lycopene	SNL-induced neuropathic pain in laboratory mice; lycopene administered intrathecally at a dose of 10 nmol.	Repeated lycopene administration prevented the occurrence of mechanical hypersensitivity.	Zhang et al. (2016) [45]
Lycopene	STZ-induced neuropathic pain in laboratory mice; lycopene administered orally at a dose of 4 mg/kg body weight for 4 weeks.	Significant alleviation of thermal hyperalgesia and cold allodynia.	Kuhad and Chopra (2008) [46]
Lycopene	STZ-induced neuropathic pain in laboratory mice; lycopene administered orally at doses of 1, 2 and 4 mg/kg body weight for 4 weeks.	Significant alleviation of thermal hyperalgesia. Reduced TNF-α and NO release in a dose-dependent manner	Kuhad et al. (2008) [47]

## 3. Carotenoids in Cancer Cachexia

Cancer cachexia was defined in 2011 by international consensus statement as “a multifactorial syndrome characterized by ongoing loss of skeletal muscle mass, with or without loss of fat mass, that cannot be fully reversed by conventional nutritional support and leads to a progressive functional impairment” [50]. As the main clinical presentation of cancer cachexia is an unintentional loss of weight; the diagnosis criteria include weight loss of more than 5% in the previous 6 months or more than 2% in the presence of BMI < 20 or sarcopenia [50]. Data obtained from palliative care settings proved that the incidence of cachexia in cancer patients ranges from 40% (in patients with breast cancer and leukemia) to 80% (in patients with gastric and pancreatic cancers), increases towards the end of life, regardless of tumor location [51]. Cancer cachexia is accompanied by a set of symptoms of the same pathophysiology, such as anorexia (loss of appetite), sarcopenia (loss of skeletal muscle tissue), and asthenia (reduced muscular strength and fatigue) [52], which are sometimes combined into one “anorexia-cachexia-asthenia syndrome”. Cancer cachexia not only induces progressive functional impairment, but also negatively affects a patient’s quality of life and responsiveness to chemotherapy, and causes a dramatic change in appearance and distress both to the patient and their caregivers [53,54]. Cancer cachexia not only limits therapeutic options for cancer patients and is associated with high healthcare costs concerning but is also recognized as a direct cause of at least 20% of cancer-associated deaths [52]. It is more of a process that, if left untreated, leads from precachexia—a partially reversible condition—to refractory cachexia with fatal outcome.

A systemic inflammation is the main pathogenetic factor of cancer cachexia, which affects the function of several tissues such as skeletal muscle, fat tissue, liver, and brain [52], leading to a negative protein and energy balance that results from a variable combination of reduced food intake, metabolic disorders such as elevated resting energy expenditure, resistance to anabolism factors, excessive catabolism with protein degradation, and insulin resistance [50,53]. Atrophy of skeletal and cardiac muscle, adipose tissue browning, above-mentioned metabolic disorders, anorexia, and fatigue are induced by inflammatory cytokines produced by the cancer cells, the stromal cells within the tumor environment, and by host immune cells as well. Cancer cells are able to secrete a variety of chemical compounds inducing catabolism of muscles and adipose tissue, such as pro-inflammatory cytokines, eicosanoids, heat shock proteins 70 (HSP70) and 90 (HSP90), activins, myostatin, transforming growth factor-β (TGFβ), and adrenomedullin [53]. An excessively increased inflammation state generated by tumor cells in a vicious circle participates in the release of pro-inflammatory factors by cells of the host immune system that are classified as procachectic factors: TNF-α, interleukins IL-1, IL-6, IL-8, and interferon gamma (IFNγ) [53]. One of the abovementioned factors, TNF-α, was initially termed “cachectin” due to its direct catabolic effect on skeletal muscle [52]. Expression of anti-inflammatory cytokines, such as IL-4, IL-10, and IL-12 remains decreased in cancer cachexia, leading to an increase in the imbalance between pro- and anti-inflammatory factors [52]. 

The most effective way to treat or slow down the progression of cancer cachexia is optimal anti-cancer therapy. However, in many patients with advanced neoplastic disease under palliative care, that procedure has been terminated earlier by the oncologists, mainly due to its ineffectiveness, side effects, or disease progression. In these cases, the symptomatic treatment of cancer cachexia should be aimed at improving quality of life without creating additional disadvantages unacceptable to the patient. Unfortunately, mentioned above, the therapeutic options are limited and, in many cases, not effective [52]

Nutrition adequate to the patient’s condition and needs, considering the appropriate feeding route, is the fundamental factor in the management of cancer cachexia. Reliable and insightful assessment of a patient’s current diet and consultation with a nutritional healthcare professional to improve the quantity and quality of food consumed is essential. A first-line approach includes oral nutritional supplements providing macro- and micronutrients for those who are unable to meet their caloric and nutritional needs by the oral route [53]. In some cases, especially in multisite gastrointestinal obstruction or malabsorption syndrome, it is necessary to intensify nutritional treatment and to introduce artificial enteral nutrition or total parenteral nutrition. However, cancer cachexia cannot be fully reversed, neither by conventional nutritional support nor even total parenteral nutrition [52]. 

Orexigenic drugs as well as drugs leading to cytokine inhibition, such as corticosteroids and progestogens (megestrol acetate) have shown positive results in clinical trials, but undesirable side effects leading to muscle atrophy and enhanced risk of thromboembolism may limit their prolonged use [53]. New therapies, such as anamorelin, a growth hormone secretagogue receptor type 1 (ghrelin receptor) agonist, and melanocortin receptor 4 antagonists that not only regulate appetite and satiety but also have systemic effects to promote protein anabolism and energy storage, are under investigation [53]. Supplementation with omega-3 fatty acids, glutamine, and branched-chain amino acids that are able to reduce IL-1 and TNF-α or to improve the efficacy of nutritional support has been also tested, but that intervention did not bring any significant benefits [55]. 

As the anabolic deficit may be partially compensated by maintaining physical activity, the introduction of a motor intervention optimized in terms of the individual’s exercise tolerance may be an important intervention improving the overall state and quality of life of patients with cancer cachexia. Especially since a systematic review of 16 randomized controlled trials (RCT) conducted in cancer patients has shown that aerobic and resistance exercises improved body muscle strength more than usual care [56]. Moreover, the abovementioned types of exercise training programs were demonstrated to have an anti-inflammatory effect [52].

Since an excessive inflammatory state underlies the disturbances leading to cancer cachexia, the inclusion of naturally occurring anti-inflammatory substances in the daily diet may be able to significantly improve the patient’s cancer-related symptoms, general condition, and prognosis. That assumption seems to be confirmed by data from the pilot randomized trial conducted by Zick et al. [57], where it was revealed that a 3-month diet rich in fruits, vegetables, whole grains, and omega-3 fatty acid-rich foods improved fatigue in breast cancer survivors. A significant increase in plasma concentrations of serum total carotenoids, β-cryptoxanthin, lutein, zeaxanthin and lycopene were observed in the study group [57]. On the other hand, studies by O’Halloran et al. [58] and Kochlik et al. [59] showed that low micronutrients status, including a low concentrations of carotenoids (lutein/zeaxanthin, α-carotene, β-carotene, lycopene, and β-cryptoxanthin) can be an easily modifiable marker and intervention target for frailty in the elderly. As frailty syndrome in the elderly and cancer-induced asthenia have many features in common, the results of those studies support the need for further research into micronutrient supplementation to prevent and relieve asthenia. 

Carotenoids are a group of naturally occurring chemical compounds whose anti-inflammatory properties have been known for centuries and have been used in folk medicine in the management of inflammation-related symptoms and diseases. Hajizadeh-Sharafabad et al. [6], in a meta-analysis of randomized controlled trials investigating the effects of carotenoids on selected inflammatory parameters, confirmed their protective role with the special ability of astaxanthin, lutein/zeaxanthin and β-cryptoxanthin to decrease C-reactive protein (CRP) levels and lycopene to reduce levels of IL-6. Kritchevsky et al. [60] performed an observational study with 4557 healthy participants; significantly lower plasma concentrations of carotenoids were associated with increased CRP levels and with high white blood cell counts [60]. 

The anti-inflammatory properties of carotenoids have been confirmed in experimental studies conducted under various conditions, both in vitro and in preclinical and clinical studies as well. Widespread access to food products containing carotenoids, as well as the increasing amount of synthetically produced carotenoids available in various forms on the market, make this group of micronutrients particularly interesting in terms of supportive treatment of cancer cachexia. Below, the authors present an overview of carotenoids with the most proven anti-inflammatory properties that may be beneficial if added to the daily diet for prophylaxis and slowing down the progress of cancer cachexia. 

### 3.1. Astaxanthin 

The studies conducted so far indicate that the anti-inflammatory properties of astaxanthin result from its ability to regulate the genes of inflammatory biomarkers, such as acute-phase proteins and abnormal activation of specific inducible enzymes, and to influence some signaling pathways [36]. As postulated in the study by Chang and Xiong [36], astaxanthin could promote the phosphoinositide 3-kinase (PI3K)/(AkT) and nuclear factor erythroid 2-related factor 2 (Nrf2) signaling pathways, while blocking NF-κB, ERK1/2, c-Jun N-terminal kinase (JNK), p38 MAPK, Janus kinase 2 (JAK2)/signal transducer. As postulated in the study by Chang and Xiong [36], astaxanthin could promote the phosphoinositide 3-kinase/protein kinase B (PI3K/AkT) and nuclear factor erythroid 2-related factor 2 (Nrf2) signaling pathways, while blocking NF-κB, ERK1/2, c-Jun N-terminal kinase (JNK), p38 MAPK, Janus kinase 2/signal transducer and activator of transcription 3 (JAK2/STAT3) signaling pathways. It can be concluded that since astaxanthin reduces excessive oxidative stress and inflammation in many chronic diseases (diabetes, gastrointestinal, neurological, hepatic and renal diseases, eye and skin disorders) [36], it may also exert a beneficial effect in patients with systemic inflammation in the course of cancer cachexia. 

In the experimental in vitro study by Choi et al. [61] astaxanthin inhibited the expression of NO, inducible nitric oxide synthase (iNOS), and COX-2, and suppressed the protein levels of iNOS and COX-2 in LPS-stimulated BV-2 microglial cells. The protective effects of astaxanthin against UV-induced inflammation was observed in a study by Yoshihisa et al. [62]. In that study pretreatment with astaxanthin caused a significant reduction in the protein content and mRNA expression of iNOS, COX-2, MIF, IL-1β, and TNF-α and decreased release of prostaglandin E2 (PGE2) from HaCaT keratinocytes after UVB or UVC irradiation [62]. 

In a preclinical study Part et al. [63] revealed that astaxanthin in a dose of 50 mg/kg body weight taken orally for 18 days provoked a downregulation of COX-2, iNOS, and intercellular adhesion molecule 1 (ICAM-1) in STZ-induced diabetic rats. In turn, in the randomized, double-blind placebo-controlled clinical trial patients with type 2 diabetes mellitus receiving 8 mg of astaxanthin orally daily for 8 weeks presented reduced plasma levels of IL-6 and MDA as well [64]. In other RCT in *Helicobacter*
*pylori*-positive patients a significant decrease in gastric inflammation and elevated CD4/CD8 response after astaxanthin supplementation (40 mg daily) was observed [65]. 

However, the most important premise confirming the possible effectiveness of astaxanthin in cancer cachexia are the results of preclinical studies conducted by Cremades et al. [54] and Nishida et al. [66]. In the first study of 48 cachectic laboratory rats—24 with tumor (AH-130 Yoshida ascites hepatoma) and 24 with no tumor—were divided into groups with 12 animals receiving a diet formulated with astaxanthin-rich crayfish enzymatic extract or a standard diet for 4 weeks [54]. The authors observed that plasma TNF-α was smaller in the crayfish enzymatic extract-treated group than in the group on a standard diet. Moreover, improved symptoms related to cancer cachexia (anorexia and weight loss both in adipose and muscle tissues) and increased survival (57.1% vs. 25.9%) were observed in the group treated with crayfish enzymatic extract. Unfortunately, no reversal of cachexia was reported [54]. In the next preclinical study mentioned above, astaxanthin treatment significantly ameliorated insulin resistance through regulation of 5′ adenosine monophosphate-activated protein kinase (AMPK) activation in the muscle and stimulated mitochondrial biogenesis, enhanced exercise tolerance and exercise-induced fatty acid metabolism, and exerted anti-inflammatory effects as well [66]. 

### 3.2. Lycopene 

Lycopene is an acyclic isomer of β-carotene, however, unlike β-carotene lycopene is devoid of the β-ionic ring structure, therefore it does not possess provitamin A activity [67]. Lycopene is one of the strongest antioxidants, with a singlet-oxygen-quenching ability twice as high as that of β-carotene and ten times higher than that of α-tocopherol [67]. Antioxidants also have anti-inflammatory properties because of their ability to interfere with proinflammatory signals and interrupt the inflammatory process. The anti-inflammatory properties of lycopene have been studied in experiments conducted in vitro and in animal models with chronic inflammation, as well as in clinical trials. 

In a study by Huang et al. [68], lycopene inhibited the binding abilities of NF-κB and stimulatory protein-1 (Sp1) to the binding sites in the matrix metalloproteinase-9 (MMP-9) promoter and reduced the expression of insulin-like growth factor-1 receptor (IGF-1R). Moreover, the decreased intracellular concentration of ROS in SK-Hep-1 cells was also observed in that study [68]. Similar results were obtained in an in vitro study by Cha et al. [69], where lycopene restrained NF-κB and JNK activation and suppressed the expression of TNF-α, IL-1β, IL-6, COX-2, and iNOS in SW480 human colorectal cancer cells. Oral administration of lycopene (10 mg/kg body weight daily) for 6 weeks significantly decreased leptin, resistin, and IL-6 gene expression in adipose tissue and plasma concentrations in laboratory rats with high-fat diet-induced obesity [70]. In a study by Fenni et al. [71], the anti-inflammatory effect of lycopene was related to a reduction in the phosphorylation levels of the nuclear factor of the kappa light polypeptide gene enhancer in B-cells inhibitor (IκB) and p65 (modulators in the NF-κB pathway) and significant decrease of obesity-related proinflammatory cytokine IL-6, chemokines (monocyte chemoattractant protein-1 (MCP-1)), MMP-9, acute-phase protein (serum amyloid A3 (SAA3)), and adipokines (leptin, visfatin) in the adipose tissue of mice on a high-fat diet. 

Burton-Freeman et al. [72] proved that a one-time consumption of lycopene may exert beneficial effects on inflammatory state and oxidation as well. They observed that tomato products significantly attenuated high-fat meal-induced increase of plasma IL-6 concentration, as well as postprandial low-density lipoprotein (LDL) oxidation in healthy weight individuals. Data from other studies presented below show the effects of chronic consumption of lycopene or lycopene-rich tomato products both in populations with a chronic inflammatory state resulting from overweight, obesity, type 2 diabetes, or heart failure, and in healthy volunteers as well.

In the RCT conducted among healthy men by Kim et al. [73] highly sensitive C-reactive protein (hs-CRP), soluble intercellular adhesion molecule 1 (sICAM-1) and soluble vascular cell adhesion molecule 1 (sVCAM-1) concentrations in plasma significantly decreased after 8 weeks of supplementation of 15 mg lycopene daily. It was also observed that changes in lycopene concentration negatively correlated with hs-CRP and superoxide dismutase (SOD) activity. Such beneficial effects were not proven in lower doses of lycopene supplementation (6 mg lycopene daily). In turn, in the study by Li et al. [74], everyday consumption of 80 mL of tomato juice (containing 32.5 mg of lycopene) for 2 months by young, healthy women resulted in significantly reduced serum levels of monocyte chemoattractant protein-1 (MCP-1) and thiobarbituric acid reactive substances (TBARS) as well (other markers of inflammation were not investigated in the study). In the RCT by Jacob et al. [75], the consumption of tomato juice containing 20.6 mg lycopene (daily dose) for 2 weeks significantly reduced CRP plasma concentrations in healthy volunteers. Unfortunately, all other inflammation markers investigated in that study (8-epi-prostaglandin F2alpha (8-epi-PGF2α), IL-1β, and TNF-α) and the antioxidant capacity of plasma and urine were reduced to a lesser extent or remained uncharged [75]. No significant effect on hs-CRP was observed in a placebo-controlled trial by Gajendagadkar et al. [76] both in healthy volunteers and patients with cardiovascular disease. No significant effect of tomato intake on insulin-like growth factor-1 (IGF-1) levels was also observed in RCT by Riso et al. [77], with healthy young subjects consuming 250 mL of a tomato drink for 26 days, separated by 26 days of wash-out. However, changes in lycopene before and after each experimental period were inversely and significantly correlated with those of IGF-1 [77]. 

Data from a study by Thies et al. [78] indicate that daily consumption of tomato-based products equivalent to 32–50 mg lycopene daily as well as supplementation of lycopene supplements (10 mg daily) are ineffective in reducing systemic inflammatory markers, such as hs-CRP, IL-6, ICAM-1 as well as oxidized low-density lipoprotein (ox-LDL) and markers of insulin resistance in overweight, healthy, middle-aged individuals. Markovits et al. [79] made similar observations in obese patients. In that interventional study 4 weeks of lycopene supplementation in tomato-derived Lyc-O-Mato (30 mg daily) did not result in reduced markers of inflammation (CRP, IL-6) and oxidation products (conjugated dienes). 

Lycopene-rich tomato juice reduced systemic inflammation in overweight and obese females consuming 330 mL/d of tomato juice for 20 days in the RCT by Ghavipour et al. [80]. Serum concentrations of IL-8 and TNF-α decreased significantly in the overweight females compared with the control group and with baseline, whereas serum IL-6 concentration was decreased in the obese intervention group compared with the control group, with no differences in IL-8 and TNF-α observed [80]. Colman-Martinez et al. [81] showed that daily consumption of 200–400 mL of tomato juice for 4 weeks resulted in significantly lower concentrations of inflammatory molecules related to atherosclerosis, such as ICAM-1 and vascular cell adhesion molecule 1 (VCAM-1), and a tendency to decrease IL-8 in a population at high cardiovascular risk. Significantly decreased plasma concentration of CRP was observed in the RCT by Biddle et al. [82] in patients with heart failure after 29.4 mg of lycopene intake per day for 30 days. Importantly, C-reactive protein levels decreased significantly in the intervention group in women, but not in men [82].

On the other hand, in the clinical trial conducted by Upritchard et al. [83], the inclusion of tomato juice (500 mL daily) in the usual diet for 4 weeks in patients with type 2 diabetes aged <75 years resulted in increased plasma lycopene levels and the resistance of LDL to oxidation, but with no changes in concentration of plasma inflammatory markers (such as CRP, VCAM-1, and ICAM-1) observed. Decreased total oxidative status at the end of the 14-day period of tomato paste supplementation (33.3 mg of lycopene daily) was observed in healthy volunteers by Xaplanteris [84] et al. An improved protection of deoxyribonucleic acid (DNA) from oxidative damage was observed by Riso et al. [85] in young healthy female subjects after 3 weeks consumption of a diet enriched with small amounts of different tomato products providing as a mean 8 mg lycopene, 0.5 mg beta-carotene and 11 mg vitamin C per day. Similar data were obtained by Porrini et al. [86] in healthy subjects who consumed 250 mL of a tomato drink daily, providing about 6 mg lycopene, 4 mg phytoene, 3 mg phytofluene, 1 mg β-carotene and 1.8 mg α-tocopherol. 

Despite the promising data obtained from preclinical studies, the results of the abovementioned clinical trials are inconsistent. Data from the systematic review of 17 intervention studies conducted by van Steenwijk et al. [67] showed that despite the increase in circulating lycopene levels after tomato or lycopene supplementation, almost no changes in inflammation biomarkers were observed. It is worth mentioning that none of analyzed studies assessed the impact of lycopene consumption on the markers of inflammation and oxidative status in cancer patients, with particular emphasis on patients with cancer cachexia. Therefore, it is still impossible to come to a conclusion on the benefits of additional supplementation of patients’ daily diet in preventing the onset and slowing down the progression of cancer cachexia. Conducting research among cancer patients might allow drawing appropriate conclusions.

### 3.3. β-Cryptoxanthin

In an in vitro study, Liu et al. [87] examined the anti-inflammatory effect of β-cryptoxanthin on LPS-induced inflammation in mouse primary Sertoli cells. It was observed that β-cryptoxanthin inhibited the LPS-induced upregulation of TNF-α, interleukin-10 (IL-10), IL-6, and IL-1β, and significantly suppressed NF-κB (p65) activation as well as p38 MAPK phosphorylation. 

Data from a preclinical study by Liu et al. [88] indicate that β-cryptoxanthin provides a beneficial effect against cigarette smoke-induced inflammation and oxidative DNA damage. β-Cryptoxanthin substantially reduced smoke-elevated TNFα in lungs and lung macrophages, activation of NF-κB, expression of activator protein-1 (AP-1), and levels of 8-hydroxy-2′-deoxyguanosine (8-OHdG) in laboratory ferrets [88]. In response to β-cryptoxanthin treatment laboratory rats in a study by Zhang et al. [89] presented decreased serum levels of TNF-α, IL-1β, and IL-6, decreased p65 expression, and activity in the nucleus and p-p38 MAPK level, suggesting that β-cryptoxanthin exerts an anti-inflammatory effect by inhibiting NF-κB-mediated inflammatory signaling. Similar data come from a study by Sahin et al. [90], showing inhibited liver and adipose tissue NF-κB and TNF-α expressions as well as upregulation of Nrf2, *heme oxygenase 1* (HO-1), peroxisome proliferator-activated receptor alpha (PPAR-α), and phosphorylated insulin receptor substrate 1 (p-IRS-1) expression after β-cryptoxanthin supplementation in laboratory rats. 

Despite the promising results of animal studies, due to the lack of data from clinical trials on the benefits of β-cryptoxanthin supplementation in preventing and slowing down the progression of cancer cachexia, no unequivocal conclusions can be made regarding beta-cryptoxanthin supplementation in palliative-care cancer patients.

## 4. Carotenoids and Cancer-Induced Fatigue 

The studies conducted so far have shown that higher fruit and vegetable intake as well as a Mediterranean diet were associated with a lower risk for frailty—a geriatric syndrome observed in the population above 65 years old. Conversely, a suboptimal consumption of vitamins A, E, D, and carotenoids was shown to be related with low physical activity, weakness, reduced muscle mass, and poor physical performance in the abovementioned population [59]. There are some common elements both in the frailty syndrome and fatigue (asthenia), one of the symptoms accompanying cancer cachexia (anorexia-cachexia-asthenia syndrome is described in the above section), in the context of the pathophysiological background and the clinical picture as well. Increased levels of inflammatory markers, excessive oxidative stress, and muscle loss or sarcopenia are closely related with the pathogenesis of both syndromes. Distinguishing between frailty syndrome in a cancer patient and asthenia observed in the course of cancer cachexia syndrome can be difficult, and in some cases, it is even impossible to do. 

In many studies the association of frailty syndrome with the elevated parameters of the inflammatory state, such as CRP, factor VII, fibrinogen, D-dimers, interleukins IL-6, IL-8 and IL-2, TNFα, and interferon gamma (IFN-γ), and with reduced albumin plasma concentration were shown [91,92,93]. Similar immune disorders are observed in cancer cachexia, hence the hypothesis that supplementing the daily diet with anti-inflammatory nutrients may reduce the frequency and intensity of symptoms associated with both fragility syndrome and cancer cachexia as well. 

The above hypothesis has been supported by data obtained from the FRAILOMIC study conducted by Kochlik et al. [59]. In that study, it was observed that robust participants had significantly higher lutein/zeaxanthin, α-carotene, β-carotene, lycopene, and β-cryptoxanthin plasma concentrations than participants with general weakness. In a systematic review performed by Zupo et al. [94] higher dietary and plasma levels of carotenoids, taken individually or cumulatively, were found to markedly reduce the odds of physical frailty.

The effect of carotenoids on exercise-related parameters was analyzed by the authors of this publication, as the inclusion of appropriate physical activity in patients with cancer cachexia and cancer-related fatigue may also be of benefit. We found the results of the study carried out by Liu et al. [95] particularly promising. In that RCT it was proven that astaxanthin supplementation enhanced metabolic adaptation with aerobic training in the elderly [95]. In another double-blind, placebo-controlled study, a formulation containing astaxanthin, tocotrienol, and zinc significantly improved muscle strength in healthy elderly in addition to an elevation in endurance and walking distance found with exercise training alone [96]. Moreover, astaxanthin supplementation in a daily dose of 4 mg was able to prevent inflammation induced by rigorous physical training in trained male soccer players with lower neutrophil count and hs-CRP level than in a placebo group [97]. On the other hand, in a clinical trial with trained athletes, markers of inflammation (hs-CRP) and exercise-induced skeletal muscle damage (creatine kinase) were equally unaffected by astaxanthin supplementation [98]. 

The authors of this article did not find in the literature data from human studies that would confirm the efficacy of astaxanthin in cancer cachexia. Despite the promising results of animal studies, no conclusions can be drawn about the effectiveness of such treatment in the population of palliative care patients. We know a little more about the possible influence of astaxanthin on the improvement of physical exertion, however, these studies were not conducted in a study group with cancer-related fatigue, so these data are also too weak to unequivocally draw a conclusion on this subject. Further research in humans should be conducted in this area.

**Table 2 nutrients-14-03183-t002:** Human studies on the effectiveness of carotenoids in alleviating cancer-related symptoms (neuropathic pain, cachexia) and frailty syndrome.

Symptom (Carotenoid)	Study Description	Main Results	References
**Neuropathic pain** **(crocin)**	Randomized, double-blind, placebo-controlled clinical trial; 171 patients (aged 25–89) with mild to severe symptomatic chemotherapy-induced peripheral neuropathy for at least one month, randomly assigned to two groups: crocin 30 mg daily p.o. vs. placebo for 8 weeks. A crossover study performed with a 2-week washout period.	Average neuropathic pain decreased significantly in the crocin group: −2.5 (54.3%) by NRS −0.8 (33.3%) by NCIC-CTC scale −0.04 (23.5%) by ECOG neuropathic scale −0.8 (47%) by WHO scale −0.4 (12.9%) by BPI −8.3 (36.2%) by McGill pain rating index −7.2 (10.8%) by SDS −0.9 (30%) by NPS compared with placebo (*p* < 0.05).	Bozorgi et al. (2021) [32]
**Pain** **(lycopene)**	Randomized clinical trial; 54 patients with metastatic prostatic cancer, randomly assigned to two groups: orchidectomy alone (27 patients) vs. orchidectomy plus lycopene (OL) 2 mg twice daily orally (27 patients). The mean (range) follow-up of the patient still alive was 25.5 (24–28 months).	11 patients in the orchidectomy and 21 in the OL group had a complete response (CR, defined as the serum PSA < 4 ng/mL and/or a normal bone scan) (*p* < 0.05). It was observed a linear relationship between the response based on the bone scan and the requirement of analgesics. Patients with CR in both groups required no analgesics, this was more expressed in the OL group (25% vs. 15%).	Ansari et Gupta (2003) [48]
**Pain** **(lycopene)**	Clinical trial; 20 consecutive patients (aged 56–90) with metastatic hormone-refractory prostate cancer; lycopene in the dose of 10 mg daily orally administered for 3 months. Bone pain present in 16 patients; 9 required nonopioid analgesics, 7 opioid analgesics.	Ten (62%) patients managed to cut down the dose of analgesics on daily basis: 6 patients from nonopioid group and 4 from opioid group (more detailed information has not been provided).	Ansari et Gupta (2004) [49]
**Cancer-related fatigue** **(total carotenoids,** **β-cryptoxanthin, lutein, zeaxanthin and lycopene)**	Pilot, randomized clinical trial; 30 breast cancer survivors, who had completed cancer treatments, were randomized: 15 receiving the diet rich in fruit, vegetables, whole grains, and omega-3 fatty acid-rich foods (FRD) and 15 receiving the control diet (GHC), for 3 months.	The intervention significantly alleviated fatigue in FRD group compared to GHC group (*p* = 0.01). Serum total carotenoids, lutein, zeaxanthin, β-cryptoxanthin, and lycopene were significantly increased.	Zick et al. (2017) [57]
**Frailty status (lutein/zeaxanthin, α-carotene, β-carotene, lycopene, and β-cryptoxanthin)**	**Cross-sectional FRAILOMIC study**; associations between plasma carotenoids and patients’ frailty status (robust, pre-frail, and frail) determined using Fried’s frailty criteria, assessed by general linear and logistic regression models. The analysis included 1450 participants (**mean age** 77.5 ± 6.5 years).	Robust participants had significantly higher lutein/zeaxanthin, α-carotene, β-carotene, lycopene, β-cryptoxanthin concentrations than frail subjects. Frail subjects were more likely to be in the lowest than in the highest tertile for α-carotene (1.69; 1.00–2.88), β-carotene (1.84; 1.13–2.99), lycopene (1.94; 1.24–3.05), lutein/zeaxanthin (3.60; 2.34–5.53), and β-cryptoxanthin (3.02; 1.95–4.69) than robust subjects.	Kochlik et al. (2019) [59].
**Frailty syndrome (total carotenoids, α-carotene, β-carotene, lutein, lycopene, β-cryptoxanthin)**	Systematic review; to evaluate the association between selected single or total carotenoids and frailty syndrome; a total of 11 trials with 27 792 participants (aged 20–94) were included in qualitative synthesis.	Higher dietary and plasma levels of carotenoids, taken individually or cumulatively, were found to reduce the odds of physical frailty syndrome.	Zupo et al. (2022) [94]

NRS—Numerical Rating Scale; NCIC-CTC—National Cancer Institute of Canada—Common Toxicity Criteria; ECOG—Eastern Cooperative Oncology Group; WHO—World Health Organization; BPI—Brief Pain Inventory; SDS—Symptom Distress Scale; NPS—Neuropathic Pain Scale.

## 5. Carotenoid Supplementation in Palliative Care Patients—Is It Worth It? Summary

There is a great interest in natural compounds used in the treatment of many symptoms and diseases nowadays. Palliative care patients with burdensome symptoms related to advanced-stage cancers, disqualified from anti-cancer treatment, are especially interested in the use of dietary supplements and herbal remedies to improve prognosis, to reduce symptom intensity, and to ameliorate quality of life. Carotenoids seem to be a group of natural chemical compounds with particularly promising properties, mainly due to their strong antioxidant, anti-inflammatory, and neuroprotective properties (Figure 2). They are also a group of substances that have been used in folk medicine for centuries to treat various diseases and to alleviate the symptoms associated with them. 

One of the particularly bothersome symptoms observed in palliative care patients with advanced cancers is neuropathic pain. Due to the unsatisfactory effectiveness of treatment of neuropathic pain with analgesics and adjuvant drugs whose antinociceptive properties have been proved in clinical trials so far, the search for new substances with analgesic properties is still at the center of interest of both researchers and practicing physicians. 

Despite the promising results of studies on the analgesic properties of carotenoids that have been conducted so far in vitro and in vivo on animal models, it is impossible to unequivocally confirm the usefulness of carotenoids in the treatment of neuropathic pain in cancer patients. As was shown above, crocetin/crocin, xanthophylls (astaxanthin, beta-cryptoxanthin, and fucoxanthin), as well as bixin and lycopene have been demonstrated to attenuate neuropathic pain in animal models. However, the authors identified only one RCT [32] that positively evaluated the effectiveness of crocin in chemotherapy-induced peripheral neuropathy, and two interventional studies that confirmed beneficial effects of lycopene in the adjuvant treatment of cancer-related pain in patients with metastatic prostate cancer [48,49]. Unfortunately, the results of the abovementioned studies, due to their small number and the lack of detailed data allowing for a thorough assessment of the analgesic properties of lycopene, do not allow for the presentation of recommendations regarding the effectiveness of crocetin and lycopene in relieving neuropathic pain. Details of human studies on the effectiveness of carotenoids in relieving cancer-related pain are presented in Table 2.

Further clinical trials should be conducted, first of all on the effectiveness of carotenoids as co-analgesics that reduce pain intensity when taken simultaneously with opioids and reduce the need for opioids. It is well known that opioids such as morphine, are highly effective and recommended in the treatment of neuropathic pain. However, the tolerance and side effects observed with its chronic administration may limit its use in clinical practice [12,99]. Safakhah et al. [99] found that the co-injection of morphine and crocin (15 and 30 mg/kg) enhanced the analgesic effect of morphine in CCI–induced neuropathic pain. In the animal model of formalin-induced pain, crocin administered i.p. at a dose of 100 mg/kg increased morphine-induced antinociception [27]. 

Cancer cachexia, with coexisting cancer-related fatigue, is the second symptom with limited therapeutic options that causes enormous distress in palliative-care cancer patients and their caregivers. Considering the pathogenesis of neoplastic cachexia and the proinflammatory factors involved, in this case carotenoids also seem to be a beneficial treatment, due to their strong antioxidant and anti-inflammatory properties that have been proven in clinical trials conducted so far. 

And again, despite the promising data obtained from preclinical studies which have demonstrated the anti-inflammatory effect of selected carotenoids (astaxanthin, lycopene, and beta-cryptoxanthin), the results of the clinical trials evaluating the effect of carotenoid consumption on the concentration of inflammatory markers presented in this review are inconsistent. Moreover, the abovementioned studies may not reflect the effect of carotenoids on systemic inflammation in cancer patients. Most clinical trials that investigate the anti-inflammatory properties of carotenoids involve young, healthy people or those with high cardiovascular risk or type 2 diabetes. In these study groups a beneficial effect of both astaxanthin, lycopene, and beta-cryptoxanthin on markers of inflammation was observed. It is worth mentioning that none of the analyzed studies assessed the impact of carotenoid supplementation on the markers of inflammation in cancer patients, with particular emphasis on patients with cancer cachexia. In patients with advanced neoplastic disease, generalized inflammation may be much more severe (and it probably is, considering the CRP levels observed in palliative care patients), and the beneficial anti-inflammatory effect of carotenoids may be weaker or nonexistent. On the other hand, there is no data on the absorption and bioavailability of carotenoids in patients with cancer cachexia. Without high-quality data from clinical trials in palliative care patients, no meaningful conclusions can be drawn. Keeping in mind the fact that prevention of micronutrient deficiency may counteract progressive deterioration in functioning, the inclusion of fruits and vegetables in the patient’s daily diet, in an acceptable form, may be a part of a potential multilevel intervention to prevent and better manage a patient’s disability connected to anorexia-cachexia-asthenia syndrome [94]. Details of human studies on the effectiveness of carotenoids in relieving cancer-related fatigue and a similar frailty syndrome were presented in Table 2. 

There are not any recommendations for daily supplementation of carotenoids available for cancer patients. Because of the wide variety in bioavailability among various carotenoids and individual host factors depending on age, sex, diseases, and genetics, such dietary intake recommendations are difficult to establish [100]. Therefore, the only recommendation should be a high intake of a variety of fruits and vegetables [100]. Doctors should be aware that many cancer patients use various dietary supplements. For example, it has been shown that one-third of men with prostate cancer take some form of dietary supplementation after diagnosis, and most of them use supplements without informing their doctors [101]. Among the factors increasing the higher use of supplements, the most common are higher income, being better educated, following a healthy lifestyle, more advanced disease, a lower quality of life, and a lower satisfaction with health status and treatment [101]. However, patients should be advised against excessive supplementation, especially of artificial dietary supplements, which in some cases may also have harmful effects on health. An example of such a harmful effect may be the results obtained in the ATBC trial and the CARET study, in which lung cancer rate and overall mortality were significantly increased in the smokers who supplemented β-carotene [102,103]. Data from the literature also suggest a relationship between hypercarotenemia and anorexia, with particular emphasis on β-carotene and, to a lesser extent, α-carotene, and zeaxanthin/lutein [104].

To sum up, despite the promising data obtained from studies conducted on animal models of neuropathic pain, the number of good-quality clinical trials confirming the analgesic properties of carotenoids in cancer-related neuropathic pain is lacking. Thus, it is impossible to establish the usefulness of carotenoids as the adjuvant therapy for neuropathic and other types of pain in palliative-care cancer patients. Similarly, despite the promising data obtained from in vitro and preclinical studies confirming the anti-inflammatory effect of selected carotenoids, the results of clinical trials evaluating the effect of carotenoid consumption on inflammatory markers are inconsistent. So far, no clinical trials have evaluated the effect of carotenoid consumption on inflammatory markers in cancer patients. Moreover, there is a lack of good-quality clinical trials confirming the effectiveness of dietary carotenoids in alleviating or slowing down cancer cachexia and related symptoms such as anorexia and asthenia. Therefore, it is impossible to establish the usefulness of carotenoids as the adjuvant therapy for cancer-related cachexia in palliative care. However, the inclusion of carotenoid-rich fruits and vegetables in the daily diet in a form acceptable to the patient may be a part of a multilevel nutritional intervention to prevent and better manage disability connected with anorexia-cachexia-asthenia syndrome. Table 3 summarizes the conclusions regarding the carotenoid supplementation in the adjuvant treatment of cancer-related symptoms discussed in the manuscript.

## Figures and Tables

**Figure 1 nutrients-14-03183-f001:**
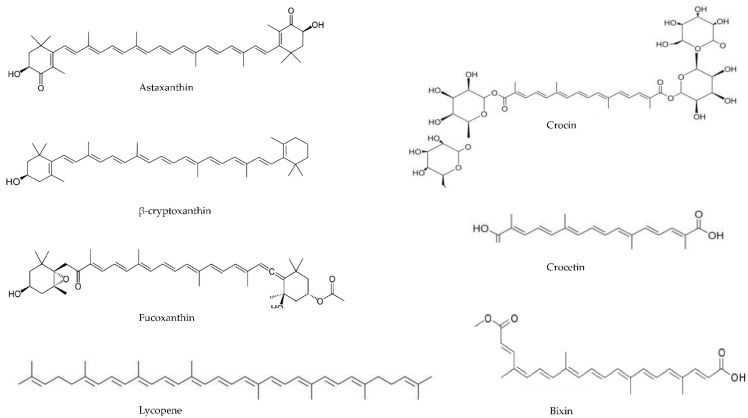
The chemical structures of key carotenoids discussed in the manuscript.

**Figure 2 nutrients-14-03183-f002:**
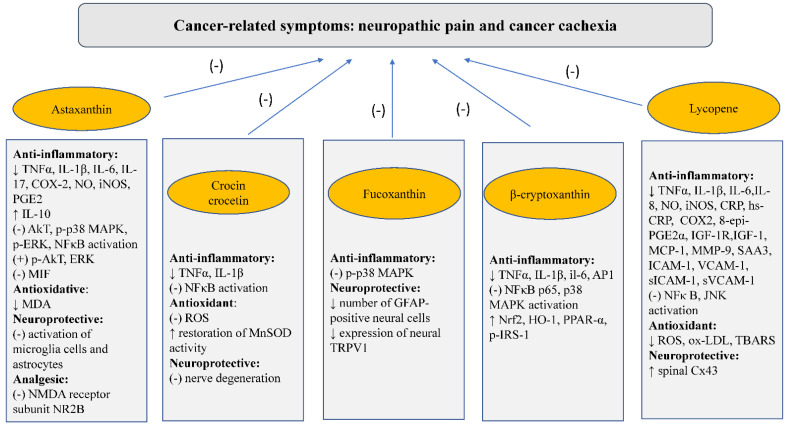
Mechanisms of carotenoids capable of relieving cancer symptoms: neuropathic pain and cachexia data obtained on the basis of the literature (↓—reduction, ↑—increase, (−)—inhibition, (+)—intensification).

**Table 3 nutrients-14-03183-t003:** Supplementation of carotenoids as the adjuvant treatment of cancer-related symptoms in palliative care-summary.

Is There Any Benefit from Supplementation of Carotenoids in the Treatment of Cancer-Related Symptoms?
Despite the promising data obtained from studies conducted on animal models of neuropathic pain, the number of good-quality clinical trials confirming the analgesic properties of carotenoids in cancer-related pain is minimal. Despite the promising data obtained from in vitro and preclinical studies confirming the anti-inflammatory effect of selected carotenoids, the results of clinical trials evaluating the effect of carotenoid consumption on inflammatory markers are inconsistent. So far, no clinical trials evaluated the effect of carotenoid consumption on the inflammatory markers in cancer patients. There is a lack of good-quality clinical trials confirming the effectiveness of dietary carotenoids in alleviating or slowing down cancer cachexia and related symptoms such as anorexia and asthenia. Considering the above-mentioned statements, it is impossible to establish the usefulness of carotenoids as the adjuvant therapy for neuropathic pain and cancer-related cachexia in palliative care. The inclusion of carotenoids-rich fruit and vegetables in the patient’s daily diet in the form acceptable for patient may be a part of multilevel nutritional intervention to prevent and better manage disability connected with anorexia-cachexia-asthenia syndrome.

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
