# Peer review of "Carotenoids in Palliative Care—Is There Any Benefit from Carotenoid Supplementation in the Adjuvant Treatment of Cancer-Related Symptoms?"

_nutrients, 2022, doi:10.3390/nu14153183_

Round 1

Reviewer 1 Report

This review can arouse readers' interest, but I think the following problems need to be solved before it is published.

Q1:A summary figure should be supplemented for the section of "conclusion and outlook", which would make this manuscript more attractive.

Q2:A summary table is recommended to be supplemented. All the examples could be listed and classified in the table, which would make the presentation more clearly.

Q3:It is recommended to provide corresponding figures or tables for key examples.

Q4:There are many abbreviations in the whole manuscript, so it is best to provide a list of abbreviations.

Author Response

Dear Reviewer,

Together with the other authors of the manuscript, we have carefully read all of Your comments. We are grateful for all Your comments, thanks to which our manuscript will be more valuable and legible.

To carefully fulfill all Your requests, we have created a few new figures and tables, among others summary figure (figure 2) and summary table (Table 6). We also provided corresponding tables with details of preclinical and clinical studies discussed in the manuscript. A list of abbreviations was also provided.

We hope that our corrections to the manuscript will prove to be sufficient to obtain your positive evaluation.

Yours faithfully,

Anna Zasowska-Nowak, MD PhD with co-authors

Reviewer 2 Report

In the review manuscript, the authors had a very deep analysis of the effectiveness of carotenoids for palliative care. The content of manuscript is very informative and the discussions are very insightful and inspirational. Overall, the content of this manuscript is very intriguing and can attract researchers in the field. Besides, the manuscript is well-written and easy to follow. I recommend the manuscript to be accepted before small changes being made. 

1. It will be better if the authors can include a figure or table with the chemical structures of key carotenoids that were discussed in the manuscript.

2. There are many typos throughout the manuscript and few of them are listed below. Please check the manuscript carefully and correct all the typos.

Please unify the using of “above mentioned” and “above-mentioned”.

Page 1, line 28: the word “colloured” should be corrected.

Page 1, line 29: The expression of “C40-based carbon isoprenoid molecules” is not correct. 

Page 1, line 31: the word “amouts” should be corrected.

Page 2, line 79: the word “terapeutic” should be corrected to “therapeutic”.

Page 3, line 121: the word “numer” should be corrected.

Page 4, line 160: the word “supress” should be corrected.

Page 4, line 165: the word “stres” should be corrected.

Page 4, line 202: the word “mechnical” should be corrected.

Page 5, line 232: the word “inreased” should be corrected.

Page 9, line 427: the word “terapeutic” should be corrected to “therapeutic”.

Page 13, line 630: the word “rewiev” should be corrected to “review”.

Page 13, line 669: the word “musce” should be corrected to “muscle”.

Page 14, line 691: the word “rewiev” should be corrected to “review”.

Page 14, lines 697-699: please check and rewrite the sentence.

Page 14, line 701: the word “strenth” should be corrected.

Page 15, line 737: the word “lycpene” should be corrected.

Author Response

Dear Reviewer,

Together with the other authors of the manuscript, we have carefully read all of Your comments. We are grateful for all Your comments, thanks to which our manuscript will be more valuable and legible.

To carefully fulfill all Your requests, we have created a few new figures and tables, among others a table with the chemical structures of key carotenoids discussed in the manuscript (Figure 1). We also checked the manuscript carefully and corrected all indicated typos. Moreover, we have created a summary figure (figure 2) and a summary table (Table 6). We also provided corresponding tables with details of preclinical and clinical studies discussed in the manuscript. A list of abbreviations was also provided.

We hope our corrections to the manuscript will prove sufficient to obtain your positive evaluation.

Yours faithfully,

Anna Zasowska-Nowak, MD PhD with co-authors

Round 2

Reviewer 1 Report

Thank you for your revisions. At present, the figures and tables provided by are relatively complete and can enrich the content of the manuscript and arouse readers' interest. However, there are still some problems, such as the format of tables (three-line table), the lack of titles in Table 6, and the necessity of listing only one item in Table 4.

Author Response

Thank You for Your review.

To solve the problems You indicated, such as a three-line table or the necessity of listing only one item, I decided to create only three tables instead of the previous six.

In Table 1, I have accumulated the data from the three previous tables (Table 1, 2, and 3) and created one entitled: "Preclinical studies on the antinociceptive effect of carotenoids on animal models of neuropathic pain."

Table 2 contains human data on the effectiveness of carotenoids in alleviating cancer-related symptoms (neuropathic pain, cachexia) and frailty syndrome. This table corresponds to Table 5 in the previous version.

Table 3 is entitled: "Supplementation of carotenoids as the adjuvant treatment of cancer-related symptoms in palliative care - summary." This table corresponds to Table 6 in the previous version.

I deleted Table 4, which contained data for only one study. Detailed data from this table I presented in the text.

We hope that all corrections made the manuscript sufficient to obtain your positive evaluation.

Yours faithfully,

Anna Zasowska-Nowak, MD PhD